# Sprayable RNAi for silencing of important genes to manage red palm weevil, *Rhynchophorus ferrugineus* (Coleoptera: Curculionidae)

**Muhammad Naeem Sattar**[ID]<sup></sup>1 *, **Muhammad Nadir Naqqash**[ID]2, **Adel A. Rezk**3, **Khalid Mehmood**2, **Allah Bakhsh**4, **Hamadttu Elshafie**5, **Jameel M. Al-Khayri**[ID]3 *

1 Central Laboratories, King Faisal University, Al-Ahsa, Saudi Arabia, 2 Institute of Plant Protection, MNS-University of Agriculture Multan, Multan, Pakistan, 3 Department of Agricultural Biotechnology, College of Agriculture and Food Sciences, King Faisal University, Al-Ahsa, Saudi Arabia, 4 Center of Excellence in Molecular Biology, Punjab University, Lahore, Pakistan, 5 Date Palm Research Center of Excellence, King Faisal University, Al-Ahsa, Saudi Arabia

* mnsattar@kfu.edu.sa (MNS); jkhayri@kfu.edu.sa (JMA-K)

**Data Availability Statement:** All the experimental data has been submitted to the journal as supplementary files. dsRNA data is available under

## Abstract

The red palm weevil, *Rhynchophorus ferrugineus* (Oliver, 1970) (Coleoptera: Dryophthoridae) is the most devastating insect-pest of palm trees worldwide. Synthetic insecticides are the most preferred tool for the management of RPW. Alternatively, RNA interference (RNAi) mediated silencing of crucial genes provides reasonable control of insect pests. Recently, we have targeted four important genes; ecdysone receptor (EcR), serine carboxypeptidase (SCP), actin and chitin-binding peritrophin (CBP) in the 3$^{rd}$ and 5$^{th}$ instar larvae RPW. The results from 20 days trial showed that the survival rate of 3$^{rd}$ instar larvae fed on SCP and actin dsRNAs exhibited the lowest survival (12–68%). While, in the 5$^{th}$ instar larvae, the lowest survival rate (24%) was recorded for SCP after 20 days of incubation. Similarly, the weight of the 3$^{rd}$ and 5$^{th}$ instar larvae treated with SCP and actin was significantly reduced to 2.30–2.36 g and 4.64–4.78 g after 6 days of dsRNA exposure. The larval duration was also decreased significantly in the larvae treated with all the dsRNA treatments. The qRT-PCR results confirmed a significant suppression of the targeted genes as 90–97% and 85–93% in the 3$^{rd}$ and 5$^{th}$ instar larvae, respectively. The results suggest that the SCP and the actin genes can be promising targets to mediate RNAi-based control of RPW.

## 1. Introduction

The date palm (*Phoenix dactylifera* L.), is among the most primitive cultivated fruit trees with evidence of its cultivation dating back to 4000 B.C. It is believed to have originated in the Middle East, and from there it reached to other regions of the world, including North Africa, the Mediterranean, and South Asia. The date palm has high economic value in many countries which provides food, shelter, and revenue for millions of people [1]. Date palm is the leading

accession numbers ON756218-33 and ON843661-64 in the NCBI GenBank database.

**Funding:** This project was financially supported King Abdulaziz City for Science and Technology (KACST), Saudi Arabia through the Project No. 11BIO1803-06.

**Competing interests:** The authors declare no conflict of interests.

fruit crop in the Kingdom of Saudi Arabia, with ~400 cultivars grown on an estimated 152 thousand hectares [2]. Saudi Arabia is the fourth top date fruit producer, with an annual production of 1.56 million tons [3]. Unfortunately, date palm trees are under the continuous threat of red palm weevil (RPW), *Rhynchophorus ferrugineus* (Oliver, 1970) (Coleoptera: Dryophthoridae). The RPW has been reported as an invasive insect pest that has caused higher yield losses to palm, including date palm, in many countries of the world [1–5]. Female weevils lay their eggs inside palm trees. After hatching within 3–5 days, the growing larvae feed on the terminal bud tissues and later the inner soft tissues and make tunnels up to a meter long and large cavities, causing significant damage to the plant [5–8].

Over the years, the commonly practiced approaches to control RPW, i.e., insecticides, entomopathogens or fumigants, soil/crown drenching, wound dressing, and use of pheromone traps could not provide a reasonable control [9]. The efficacy of several insecticides using different application methods, either alone or as combination is also questioned to control RPW [10–12]. Furthermore, the excessive and continuous use of synthetic insecticides may pose secondary hazards, including the resistance development, human and environmental health hazards [13,14]. Insects use various detoxifying enzymes to hydrolyze insecticides as a crucial biological mechanism against several synthetic insecticides [15]. Thus, detoxifying exogenously applied compounds' metabolism may be the primary cause of ineffective chemical insecticides [16,17]. Therefore, adopting environmental-friendly strategies, such as molecular approaches, is direly needed sustainably manage RPW. The RNA interference (RNAi) has the potential to promote the development of novel approaches toward plant protection sustainably. Numerous successful examples of gene silencing through RNAi against different groups of insects have been reported, such as in coleoptera, diptera, hemiptera, and lepidoptera [18–23].

Ecdysone receptor (EcR) plays a crucial role in insect molting, metamorphosis, and reproduction [24]. Serine carboxypeptidase (SCP) belongs to the serine protease family and is involved in body development, reproduction, juvenility and host interaction in the plant parasitic nematodes and pathogens [25,26]. Actin proteins are essential cytoskeletal proteins in cellular components and have been extensively studied as an RNAi target in many insect pests, where they cause lethal aberrations [27]. Actin is a very important protein that plays a role in many different cellular processes. It is essential for cell motility, cell division, muscle contraction, cell signaling, and endocytosis [28,29]. Chitin is a long-chain polymer of N-acetylglucosamine, and it is the key component of the insect exoskeleton. The chitin-binding peritrophins (CBPs) family of proteins bind to chitin, and they play a role in the formation, maintenance, and regulation of these extracellular structures [30,31]. Therefore, this study focused on the enzymes involved in insect development, such as EcR, SCP, actin, and CBP, for developing a safer control strategy at the molecular level through RNAi against this important insect pest. Besides, biological parameters like larval weight gain, survival and development were also studied.

## 2. Materials and methods

### 2.1. Red palm weevil population

Initially, infested date palm trees were inspected in the field and the RPW population (larva, pupa, and adult) was collected from date palm orchards in Bahawalpur (GPS location: 29.1342497089719, 71.47904164349954). The field-collected RPW individuals were carried to the insect-rearing laboratory at the Muhammad Nawaz Shareef University of Agriculture Multan (MNSUAM). Adults were housed in 1 kg plastic boxes and were fed on 10% (w/v) sugar solution via soaked cotton in the sugar solution. Eggs were separated and placed in Petri dishes

**Table 1. List of primers used for qRT PCR analysis of CPB, Actin, SCG and ERC gene transcripts.**

| Primer | Sequences | Target gene |
|---|---|---|
| ECR-F | AGCTACGACCCCTACAGTCC | *Ecdysone Receptor* |
| ECR-R | CTTCCTGTTGGCGTGGAGTT | |
| SCP-F | ATGCCAACACCAGAAACACG | *Sec23* |
| SCP-R | CCTCGGAAATCAAACCGACC | |
| Actin-F | GGTCGTACCACCGGTATTGTC | *Actin* |
| Actin-R | AAGTCACGACCAGCCAAGTC | |
| CBP-F | TGACTGCTCAGCCTTCTACG | *Chitin binding protein* |
| CBP-R | AGGATTAGGACGGGTACCACA | |

(diameter: 8.00 cm; height: 2.50 cm) containing moist filter paper. After egg hatching, larvae were transferred to 1 kg plastic boxes and reared on a semisynthetic artificial diet to acquire F1 progeny for future experimentation [32]. The RPW population was maintained under controlled environmental conditions at 25°C ± 1, 12 h light-dark photoperiod and 70 ± 5% relative humidity.

## 2.2. Target genes selection, synthesis and cloning

Four genes, viz. EcR, SCP, actin and CBP were fully synthesized using Macrogen gene synthesis services (Macrogen Korea). All synthesized genes were cloned into pMG-Kan cloning vector (Macrogen, Korea) with selection against kanamycin. The resultant recombinant plasmids were pMG-EcR, pMG-SCP, pMG-Actin and pMG-CPB, respectively. All the recombinant plasmids were further transformed into *E. coli* cells to facilitate steady transformation.

After multiplication, the respective gene segments were further cloned separately into the RNAi cloning vector L4440 at Bgl-II and Hind-III restriction endonucleases (ThermoFisher Scientific) following Kamath and Ahringer [33]. Each construct was confirmed by PCR amplification using the respective primers for each gene (Table 1).

All the RNAi constructs were further confirmed through Sanger sequencing at Macrogen, Korea. The constructs were re-named as L4-EcR, L4-SCP, L4-Actn, and L4-CBP, respectively (Fig 1).

## 2.3. dsRNA synthesis in bacteria

The recombinant L4440 plasmids carrying the targeted gene fragments were transformed into chemically competent HT115 (DE3) cells following Zhu et al. [34]. The HT115 transformants (carrying the recombinant plasmid) were selected on 25 mg/mL ampicillin in an overnight culture. The diluted (100X) cultures were further grown to reach an OD600 = 0.4. Later, IPTG (10 mM) was added in the freshly prepared cultures to 1 mM concentration. Finally, the cultures were incubated at 37°C for about 5 h on regular shaking. Ultimately, the culture was given a 20 min heat shock at 80°C and stored at -20°C.

## 2.4. Identification of dsRNA in bacteria

Total RNA was extracted from bacteria using TRIzol reagent to analyze dsRNA production. DNase-I was used on the isolated total RNA to eliminate any remaining DNA. Concentration was determined by loading 4 g of extracted RNA onto a 1% agarose TBE gel and staining it with ethidium bromide. The pellet was resuspended in 50 L of double-distilled water. Zhu et al. [34] was followed to normalize the sample.

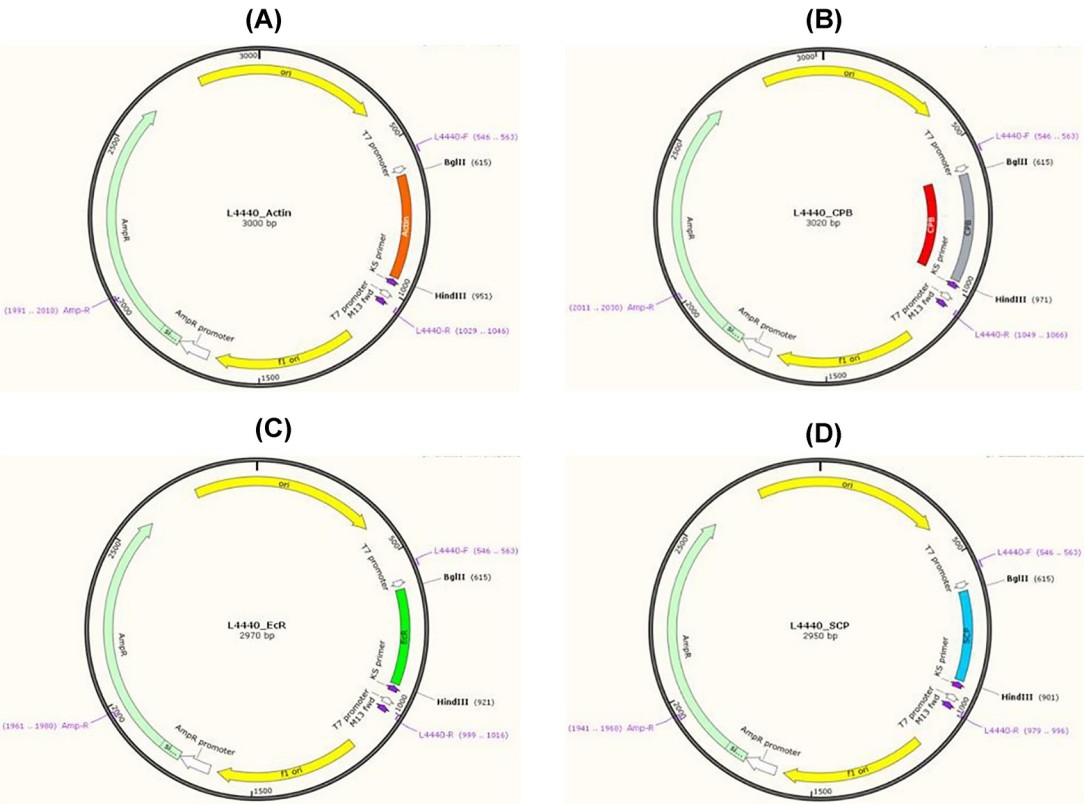

**Fig 1.** The fragments of actin (A), CPB (B), EcR (C) and SCP (D) genes of RPW were cloned in L4440 vector and were named as L4440_Actin, L4440_CPB, L4440_EcR and L4440_SCP. The maps were developed using SnapGENE.

## 2.5. Biological studies to assess the impact of dsRNA

Using the diet integration feeding approach, a bioassay was conducted on RPW larvae in the 3rd and 5th instars using dsRNA following the modified methodology of Al-Ayedh et al. [35]. The semi-synthetic diet carrying the bacterial transformants for each studied gene was prepared according to Aldawood et al. [32] and cut into 3 cm$^3$ pellets. For the feeding bioassay, 2 µg of each dsRNA was separately incorporated into the diet, followed by pelleting and the pellets were kept independently in cups. After 24 h starvation, the larvae were allowed to feed on dsRNA-treated pellets. The experiment lasted for six days and the diet was continuously replaced with a fresh dsRNA-treated diet every 24 h after confirming that larvae were feeding on the provided diet. The control group was fed only the diet treated with an empty vector. The experiment was performed with five replicates in each treatment group, and each replicate included 5 individual larvae. Larval mortality was recorded after 3 and 6 days. After preliminary feeding bioassay, the effects of a decreased dose of dsRNAs (1 µg) were used to study the larval survival, their development duration and final weight gain under laboratory conditions. Pupal duration and adult emergence from the pupae of treated larvae was also calculated. For this purpose, after 6 days, the larvae were transferred to an untreated diet for pupation. The experiments were conducted in a controlled environment with a temperature of 25 ± 1°C, humidity of 70 ± 5%, and a 12:12 light-dark photoperiod. The biological parameters, including mortality, survival, developmental period, and final weight gain/lost, were measured for each group according to the methodology of Naqqash et al. [36]. For weight gain/loss, the RPW larvae were weighed using a sensitive weighing balance (Model: ATX224; SHIMADZU) before

the feeding of the dsRNA-treated diet and then weighed after the 3$^{rd}$ and 6th day of the feeding of dsRNA treated diet. The change in weight was calculated by subtracting the initial weight (IW) from the final weight (FW) as:

$$WG = FW - IW$$

## 2.6. Validation of dsRNA effect through qRT-PCR

Using qRT-PCR, the level of expression for each gene transcript was validated. The primer-BLAST tool in NCBI (https://www.ncbi.nlm.nih.gov/tools/primer-blast/) was used to design PCR primers for targeted genes. Total RNA was extracted from a set of test larvae (3 larvae pooled in each set) as described above after seven days of treatment. First-strand cDNA was generated from 1 µg of RNA using oligo dT primers and MMLV reverse transcriptase (Thermofisher). The real-time PCR detection system (CFX-96 Touch™, BioRad, USA) was used to perform qRT-PCR assays. The qRT-PCR reactions were prepared using 50 ng cDNA, 0.5 µM of each gene specific primers, 10 µl of SYBR Green Master Mix (BioRad, USA) and nuclease-free water to a final volume of 20 µl. The universally expressed tubulin gene of RPW was used as a housekeeping gene to normalize the expression levels of individual genes in the experiment. The relative expression of the targeted genes was calculated following Livak and Schmittgen [37]. All qRT-PCR assay was repeated twice with three biological replicates.

## 2.7. Statistical analysis

The data were analyzed using a one-way ANOVA test with a significance level of 0.05 ($P \leq 0.05$). Tukey's multiple comparison test was then used to find the difference between treatments ($P \leq 0.05$). While, Kaplan-Meier method was used to analyze survival data.

# 3. Results

## 3.1. Effect of dsRNAs on mortality of larval stages of RPW

The effects of feeding two larval instars of the RPW with dsRNAs targeting ECR, SCP, actin, and CBP were investigated. Table 2 shows the percent mortality of the third instar RPW larvae fed on four different dsRNAs. There was a significant difference in mortality rates between the four groups ($P \leq 0.05$). As dsRNA works slowly, data were recorded 3 and 6 days after treatment (DAT). At 3 DAT, higher mortality rate (53.19±3.86%) was observed in the larvae fed on

**Table 2. Measurement of the mortality rate of RPW third and fifth instar larvae after three and six days of exposure to three different dsRNAs.**

| Treatments | Percent Mortality (Mean±SE*) | | | |
|---|---|---|---|---|
| | Third instar larvae | | Fifth instar larvae | |
| | 3 DAT** | 6 DAT | 3 DAT | 6 DAT |
| ECR | 37.81±6.05b*** | 65.00±4.87b | 33.95±3.80b | 49.33±4.73b |
| SCP | 45.47±4.73ab | 81.48±4.56ab | 41.67±7.17ab | 61.15±7.35ab |
| Actin | 53.19±3.86a | 86.05±5.58a | 49.33±4.73a | 65.12±7.85a |
| CBP | 37.81±6.05ab | 68.98±4.87b | 33.95±3.80b | 53.19±3.86ab |
| Control | 0.00±0.00c | 3.75±3.75c | 0.00±0.00c | 0.00±0.00c |

* SE = Standard Error Degree.

**Mean values followed by the different letter in the same column are statistically different ($P \leq 0.05$).

*** DAT = Day after treatment.

actin-dsRNA. The mortality from ECR and CBP-dsRNA was the lowest (37.81 ± 6.05%) among the treatments. At 6 DAT, the mortality of third instar larvae fed on different dsRNAs was increased in parallel to the increasing time of feeding. The mortality rate was higher, i.e., 86.05±5.58%, in actin-fed third instar RPW larvae. While the lowest mortality was found in for ECR, viz. 65.00±4.87%. No mortality was observed in the control treatments at 3 and 6 DAT (P ≤ 0.05).

Similarly, in the fifth instar larvae, a significant difference in mortalities was observed caused by the four tested genes (P ≤ 0.05). The larvae fed on actin-dsRNA showed the highest mortality (49.33±4.73%), while the lowest mortality (33.95±3.80) was observed due to ECR-dsRNA at 3 DAT (Table 2). Meanwhile, the mortality rate of the fifth instar RPW larvae was highest, i.e., 65.12±7.85% in actin-fed fifth instar RPW larvae and the lowest mortality was found in ECR, viz. 49.33±4.73% at 6 DAT. Comparatively, in the control treatments, no mortality was observed at 3 and 6 DAT (P ≤ 0.05).

## 3.2. Effects of dsRNAs on survival of two larval stages of RPW

The survival rate of 3rd instar larvae fed on dsRNA-treated diet differed significantly between the five treatments (P ≤ 0.05). After 5 days, the control treatment has a significantly higher survival rate (100%) than the other treatments. While significantly lower survival rates were observed in RPW larvae fed on SCP (60%) and actin (68%), respectively. The survival rate of CBP and EcR ranged between 84–88%. After 10 days of incubation, the survival rates of third instar larvae treated with SCP and actin dsRNAs decreased to 40% and 48%, respectively. While the survival rate of CBP was 64% and EcR was 68%. Whereas a 100% survival rate was observed in the control treatment. After 15 days of incubation, the survival rate in both treatments (SCP and actin dsRNAs) was significantly decreased to 28%. At the end of the experiment, after 20 days of incubation, the survival rate was reduced to a significantly lowest value, i.e. 24% and 28% in SCP and actin treatments, respectively. While the control larvae depicted the highest survival rate, viz. 96% after 20 days of treatment (Fig 2A).

The survival rates of the 5th instar larvae fed on dsRNA treated diet showed no significant differences after five days of incubation with 84%, 84%, 92%, 96% and 100% survival rates in SCP, actin, CBP, ECR and control, respectively (P > 0.05). However, after 10 days of incubation, relatively lower survival rates were found in actin (64%) and SCP (68%), fol-lowed by CBP (72%) and EcR (76%), respectively, compared to the control (100%). After 10 days of incubation, the survival rates of the 5th instar larvae in the actin, SCP, CBP, and EcR groups were 44%, 48%, 52%, and 56%, respectively. After 20 days of incubation, the survival rates in these groups were 36%, 44%, 40%, and 48%, respectively. Meanwhile, the control larvae depicted the highest survival rate, viz. 96% after 20 days of incubation (Fig 2B).

## 3.3. Weight gain in different stages of RPW due to dsRNA feeding

Data regarding weight changes depicted significant differences between the treatments. After 3 and 6 days of exposure, the 3rd and 5th instar larvae showed significant weight changes (P ≤ 0.05). The initial weight of the third instar larvae ranged from 0.93 to 1.01 g. The control group had a significantly higher final weight (2.83 ± 0.05 g) than the dsR-NA-treated groups, which had significantly lower final weights (1.53–1.88 g) after 3 days of treatment. After 6 days of treatment, the larval weights of the CBP and ECR treatment groups were not significantly different from the control (2.90–3.15 g). However, the larval weights of the SCP and actin treatment groups were significantly lower (2.30–2.36 g) (Fig 3A).

The initial weight of the fifth instar larvae varied from 3.87 g to 4.15 g in all treatments. After 3 days, no significant difference was observed in the weight gain of control (4.79±0.01 g)

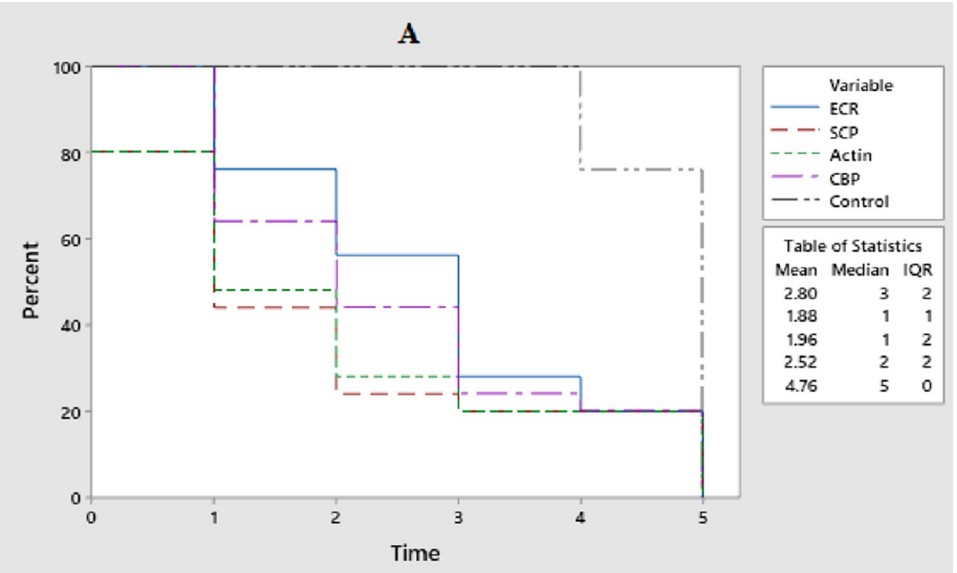

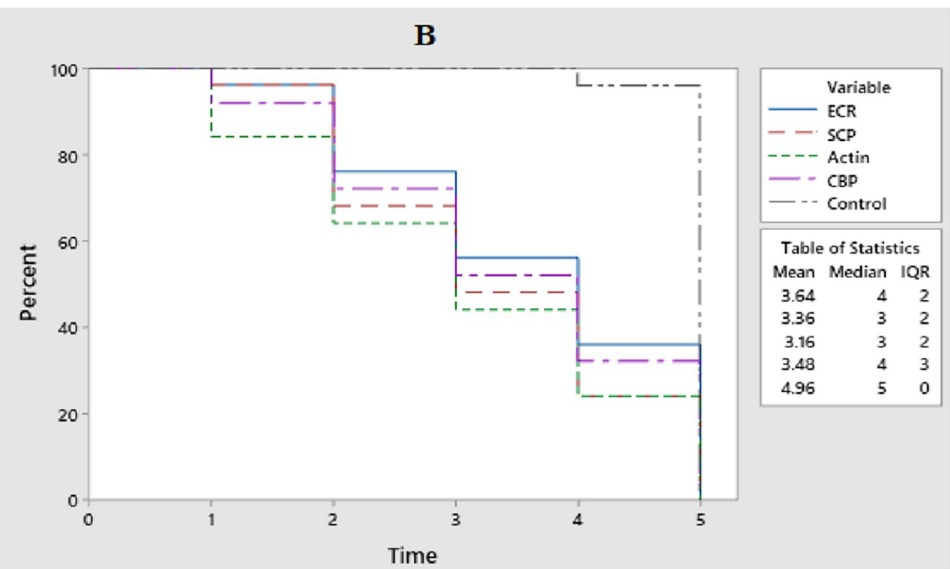

**Fig 2.** Survival of (A) 3rd instar and (B) 5th instar of RPW after feeding on different dsRNAs.

and ECR treatment (4.73±0.01 g). Whereas significantly lower weights were observed in the larvae treated with SCP, actin and CBP dsRNAs, ranging between 4.30–4.49 g (Fig 3B). After 6 days of treatment, the larval weight was significantly re-duced only in SCP and actin-treated larvae ranging between 4.64–4.78 g compared to other treatments, including control (5.24–5.65 g) (Fig 3A).

### 3.4. Effect of dsRNAs on developmental time of different larval instars of RPW

The effects of dsRNAs on larval development were measured in third instar larvae. All treatment groups showed significant differences in larval development time ($P \leq 0.05$) (Table 3). In the 3rd, 4th, 7th, 8th, 9th and 10th larval instars, all treatments had significantly shorter

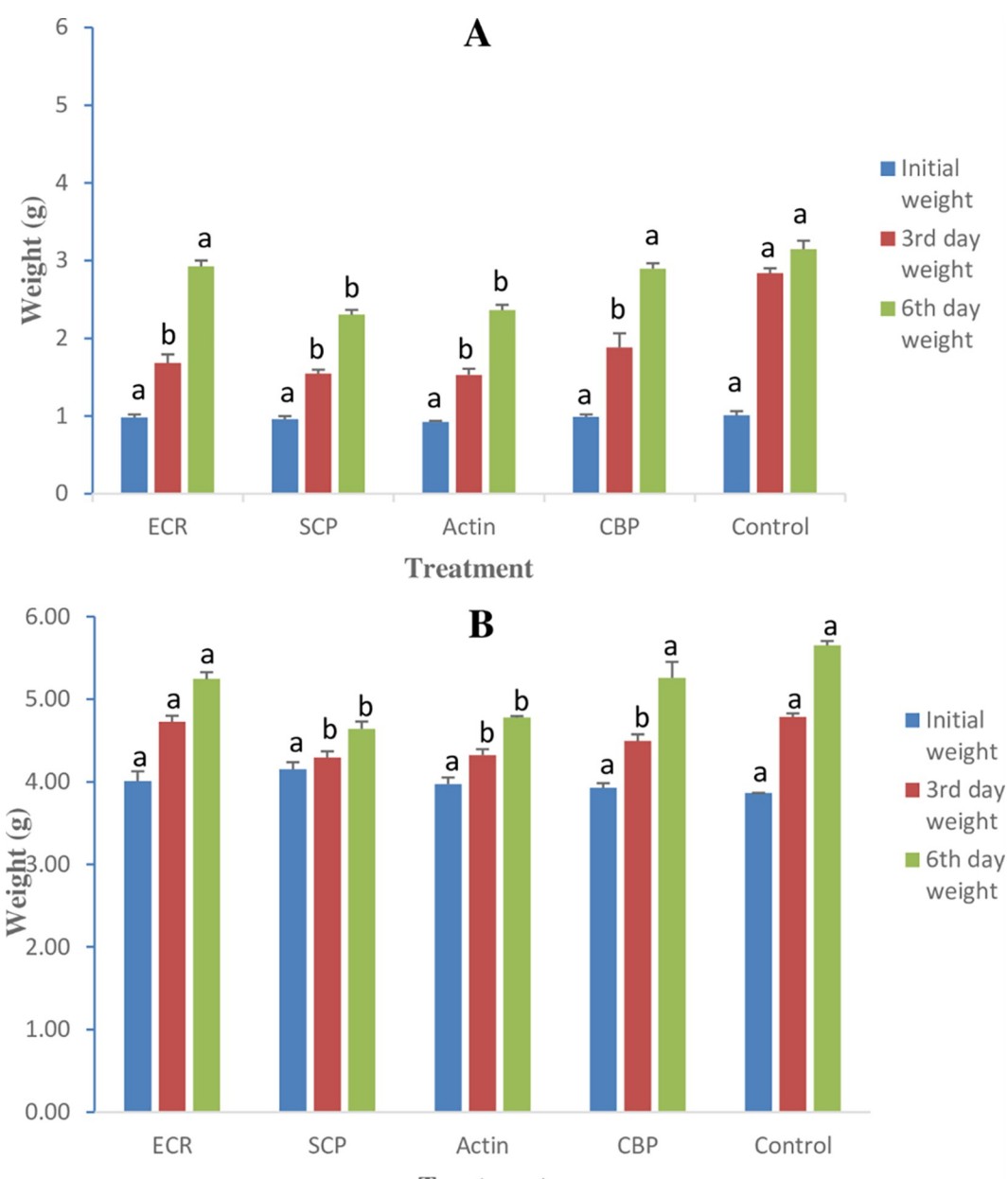

**Fig 3.** Weight change in (A) third instar and (B) fifth instar of RPW after feeding on different dsRNAs. Mean values followed by the different letter on the similar colored are statistically different (P ≤ 0.05).

durations than the control. However, there were no significant differences in duration between the dsRNA treatments (Table 3). The duration in the SCP and actin treatments was the lowest in the aforementioned larval instars. The duration of the fifth instar was significantly lower in ECR, SCP and actin treatments compared to CBP and control. In the sixth instar, SCP significantly lowered the duration period (4.60±0.24 days) than the other three treatments (5.80–4.80 days). Finally, at the 11th instar, the duration was significantly reduced in the ECR, SCP and Actin treatments compared to CBP and control (Table 3).

Similarly, all treatment groups showed significant differences in development time of 5th instar larvae (P ≤ 0.05). In the 5th, 6th, 7th and all other larval instars, all treatments had

**Table 3. Duration of different larval instars fed on different dsRNAs during 3rd instar.**

| Treatment | Larval duration (Mean±SE*) | | | | | | | | |
|---|---|---|---|---|---|---|---|---|---|
| | 3rd instar | 4th instar | fifth instar | 6th instar | 7th instar | 8th instar | 9th instar | 10th instar | 11th instar |
| ECR | 5.00±0.32b** | 4.80±0.37b | 5.00±0.32b | 5.00±0.32bc | 5.00±0.32b | 4.80±0.37b | 4.80±0.20b | 5.40±0.24b | 3.00±0.45b |
| SCP | 4.20±0.20b | 4.00±0.32b | 4.80±0.20b | 4.60±0.24c | 4.60±0.40b | 3.80±0.37b | 4.60±0.24b | 4.40±0.24b | 3.20±0.37b |
| Actin | 4.40±0.24b | 4.60±0.24b | 4.60±0.24b | 4.80±0.37bc | 4.40±0.24b | 4.40±0.40b | 4.80±0.37b | 4.80±0.37b | 3.80±0.37b |
| CBP | 5.20±0.20b | 5.00±0.45b | 5.60±0.40ab | 5.80±0.49b | 4.60±0.40b | 4.20±0.49b | 5.20±0.49b | 5.20±0.37b | 4.00±0.32ab |
| Control | 6.80±0.37a | 6.80±0.49a | 6.60±0.40a | 7.80±0.37a | 7.00±0.32a | 7.20±0.37a | 7.20±0.37a | 7.80±0.37a | 5.40±0.24a |

\* SE = Standard Error Degree.

\*\*Mean values followed by the different letter in the same column are statistically different (P ≤ 0.05).

significantly shorter durations than the control. The duration in the SCP and actin treatments was the lowest in the aforementioned larval instars. The duration of the 5th instar was lower in SCP (3.80±0.20 days) and actin (4.20±0.37 days) treatments as compared to the control (6.80 ±0.37 days). In the 6th instar, SCP and actin significantly lowered the duration period (4.00 ±0.32 days) than the control (7.00±0.45 days). However, at the 11th instar, there was no significant difference in the duration (Table 4).

## 3.5. Effect of dsRNAs on pupal duration of RPW

Pupal duration recorded in the treated 3rd instar larvae after their exposure to dsRNA for 3 days varied between the treatments (P ≤ 0.05). The pupal duration was lower in the SCP and actin treatments (12.20±0.20 and 12.00±0.32 days, respectively), whereas it was significantly higher in the control (14.00±0.32). Similarly, lower pupal duration was observed in actin (11.20±0.24 days) and SCP (11.40±0.20 days). While, the pupal duration in the control was significantly higher viz., 13.80±0.20 days (Fig 4).

## 3.6. Effect of dsRNAs on adult emergence and reproductive parameters

Adult emergence from the pupae of treated 3rd instar larvae after their exposure to dsRNA for 3 days varied between the treatments (P ≤ 0.05). Adult emergence was lower in SCP and actin treatment i.e., 8.00±4.89%, while it was significantly higher in the control (84.00±4.00%). Similarly, lower adult emergence from pupae of actin (20.00±4.89%) and SCP (24.40±4.00%) treated 5th instar larvae was observed. While, the adult emergence in the control was significantly higher viz., 88.00±4.89% (Fig 5).

**Table 4. Duration of different larval instars fed on different dsRNAs during 5th instar.**

| Treatment | Larval duration (Mean±SE*) | | | | | | |
|---|---|---|---|---|---|---|---|
| | 5th instar | 6th instar | 7th instar | 8th instar | 9th instar | 10th instar | 11th instar |
| ECR | 4.80±0.37b | 4.80±0.37b | 3.80±0.37c | 5.00±0.32b | 4.40±0.24b | 4.80±0.37b | 3.60±0.24a |
| SCP | 3.80±0.20b | 4.00±0.32b | 4.60±0.24bc | 4.40±0.24b | 4.00±0.00b | 4.00±0.32b | 3.40±0.24a |
| Actin | 4.20±0.37b | 4.00±0.32b | 4.40±0.24bc | 4.60±0.24b | 4.00±0.32b | 4.20±0.37b | 3.80±0.20a |
| CBP | 5.00±0.32b | 4.80±0.37b | 5.20±0.20b | 5.20±0.37b | 4.60±0.40b | 4.20±0.49b | 3.80±0.20a |
| Control | 6.80±0.37a | 7.00±0.45a | 7.00±0.32a | 7.60±0.40a | 7.00±0.32a | 7.40±0.40a | 4.40±0.40a |

\* SE = Standard Error Degree.

\*\*Mean values followed by the different letter in the same column are statistically different (P ≤ 0.05).

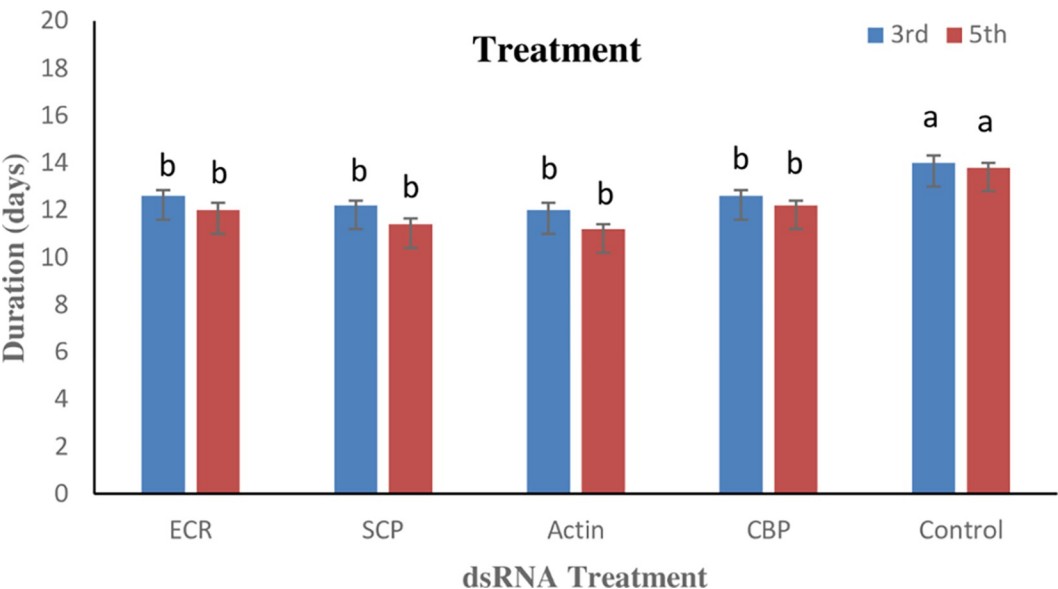

**Fig 4. Pupal duration of 3ʳᵈ instar and 5ᵗʰ instar larvae of RPW after feeding on different dsRNAs.** Mean values followed by the different letter on the bars are statistically different ($P \leq 0.05$).

There was no significant difference on egg laying capacity in control, CBP, and ECR treatments. However lower fecundity per female was observed in actin and SCP fed insects. Similarly, lower hatching efficiency, less oviposition days and prolonged pre-oviposition period was observed in these two treatments (Table 5).

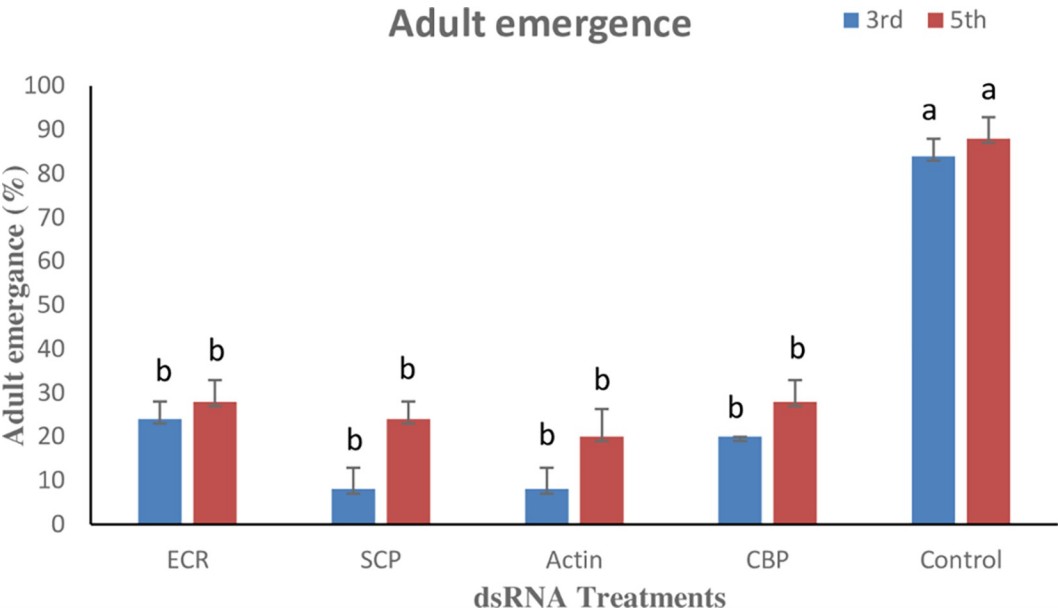

**Fig 5. Adult emergence of 3ʳᵈ instar and 5ᵗʰ instar of RPW after feeding on different dsRNAs.** Mean values followed by the different letter on the bars are statistically different ($P \leq 0.05$).

**Table 5. Sublethal effects of dsRNA on reproductive and oviposition parameters of red palm weevil.**

| Targeted gene | Number of Eggs per female (Mean±SE*) | Hatching efficiency (Mean±SE) | Oviposition Days (Mean±SE) | Pre-oviposition period (Mean±SE) |
|---|---|---|---|---|
| ECR | 152.30±6.95a** | 79.94±1.20ab | 30.90±0.81ab | 7.40±0.28c |
| SCP | 124.70±7.43b | 73.97±1.14c | 29.60±0.79ab | 8.60±0.27a |
| Actin | 130.80±7.81b | 76.03±1.35bc | 28.30±0.86b | 8.30±0.21ab |
| CBP | 150.40±6.28a | 76.56±1.25bc | 30.20±0.97ab | 7.01±0.24bc |
| Control | 159.10±7.21a | 83.43±1.79a | 32.50±0.83a | 6.90±0.95c |

* SE = Standard Error Degree.

**Mean values followed by the different letter in the same column are statistically different (P ≤ 0.05).

### 3.7. Relative expression in RPW larvae

A significant reduction was observed in the expression of all studied gene transcripts in the third instar larvae fed on the respective dsRNA-treated diet compared to controls (untreated) (Fig 6). The control group was measured separately for each gene using their individual

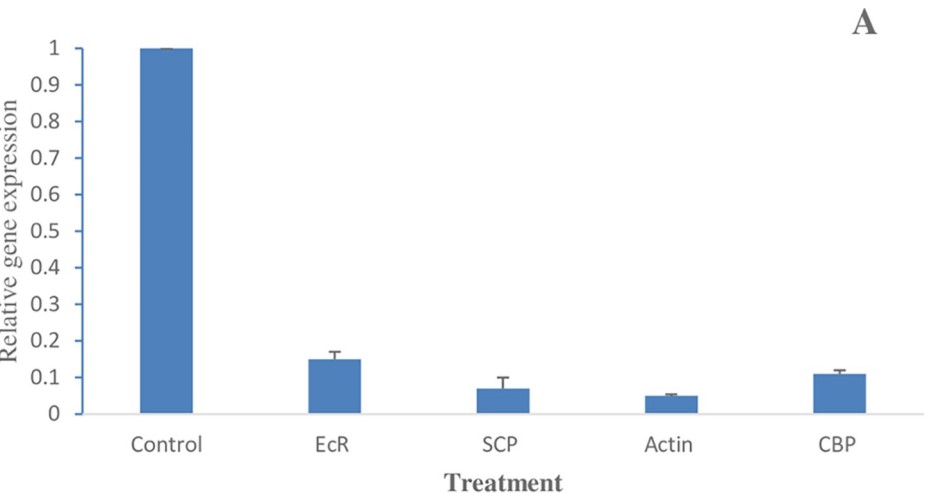

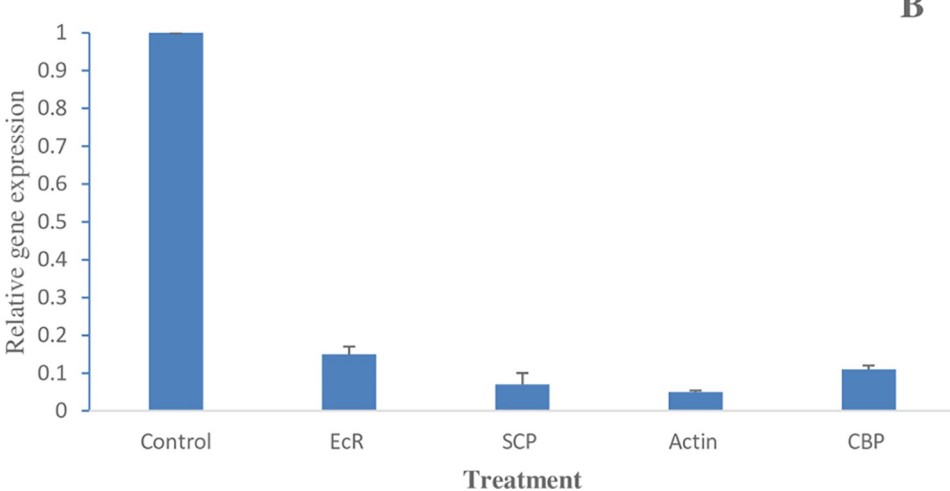

**Fig 6.** Effect of feeding dsRNAs on target-gene expression (Mean ± SE) in RPW (A) 3$^{rd}$ and 5$^{th}$ instar larvae after 6 days of feeding assay.

primers (Table 1). The expression of the control group was taken as 1.0000 in all cases. Relative gene expression in the case of different dsRNA treatments varied significantly between the treatments. The relative expression of ECR, SCP, actin, and CBP were 0.1 (90% decrease), 0.04 (96% decrease), 0.03 (97% decrease), and 0.09 (91% decrease), respectively It was found that the transcript level of SCP and actin genes was most significantly reduced compared to other treatments in the third instar larvae.

## 4. Discussion

The global spread of RPW has made it an economically important pest in date palm production. Many local or regional management practices and chemical-based pesticides, are routinely used to control RPW. However, the injudicious use of chemical insecticides has instigated specific serious problems, including insect resistance development, human health hazards, and environmental threats [13,38]. Our study showed that among four different dsRNAs treatments, the mortality rate of the third and fifth instar RPW larvae fed on actin-dsRNAs was significantly highest at 3 and 6 DAT. While, the survival rate of third and fifth instar larvae fed on SCP and Actin-dsRNAs was significantly reduced among all treatments and compared to the control after 20 days of incubation. Our results corroborate the previous studies where a significantly lower survival rate was reported in beetles due to the downregulation of SCP transcripts [39,40]. When SCP levels are low, the insect's development is compromised. This is because SCPs, such as trypsins and chymotrypsins, are the main digestive enzymes in the insect's midgut. These enzymes break down proteins into amino acids, which are essential for growth and development [25,41,42]. Previous studies have shown a direct relationship between insect survival and actin expression [27–29,43]. The actin gene is well-documented for its role in many cellular processes, including muscle contraction, cell motility, cytoskeletal structure, cell division, intracellular transport, and cell differentiation. The three main isotypes of actin (alpha, beta, and gamma) each play a specific role in these processes. For example, alpha-actin is essential for muscle contraction, while beta-actin is involved in cell motility. Gamma-actin is involved in cytoskeletal structure and cell division [29,44].

Weight gain is an important indicator of growth and development in chewing in-sects. It can also be used to assess the efficacy of ingested dsRNA [9]. In the current study, we incubated RPW larvae and analyzed their weight gain before and after the experiment. The larvae that were fed actin and SCP-dsRNAs gained significantly less weight than the control group. These findings are similar to previous research that has shown that down-regulation of actin can lead to decreased insect development [45]. In fact, weight is directly linked with actin, as in the myofibrils of muscle cells. Where actin serves as the primary contractile component and ATP synthase is the enzyme responsible for generating cellular energy [46].

The developmental time of RPW is an important factor in assessing the population growth of date palm plants. Larvae fed on dsRNA developed faster than control larvae, taking 3.80–5.40 days to reach each stage, compared to more than 6 days for control larvae. This reduction in larval developmental time has been reported previously [36,47]. Rewitz and Gilbert [48] found that the CBP protein mediates the growth hormones in insects, and that any interference in this enzyme may lead to a change in larval development. Similarly, the SCP protein is associated with cuticle differentiation in Drosophila melanogaster [49]. Qiao et al. [48] found that changes in cuticle protein can affect larval growth in silkworms. Whereas, EcR and actin signaling influence neuronal remodeling by regulating the expression of the 20E receptor. This signaling pathway may also play a role in larval development [50].

The relative change among the expression levels of targeted genes were analyzed after dsRNAs feeding. The results of qRT-PCR demonstrated a significant reduction in the

expression levels of all targeted genes in the 3$^{rd}$ and 5$^{th}$ instar larvae. The actin gene showed the most significant downregulation, followed by SCP, CPB, and EcR. The reduced expression of targeted genes established the efficacy of dsRNAs. The results are in agreement with the earlier studies [34,36].

The current study found that RPW larvae that were fed different dsRNAs at lower doses had lower expression levels of the targeted gene (s), less body weight gain, and different preadult length, weight, and survival rates. Among these dsRNAs, more significant effect on survival was observed in the actin treatment. These results are encouraging for controlling the notorious RPW pest. The study used an alternative, effective, and reliable method of developing small dsRNA molecules using bacterial expression system. This could lead to a better integrated pest management approach.

## 5. Conclusions

Red palm weevil threatens date palm production worldwide, especially in Saudi Arabia. An adequate eco-friendly management approach can help better circumvent this notorious pest. Silencing four RPW growth-related genes produced promising results in this study. The results improvise essential evidence that the survival rate, larval weight and development time of third and fifth instar larvae were significantly affected by silencing the actin and SCP genes. Future studies should focus on practicing and optimizing an appropriate delivery method for dsRNAs in insect larvae, viz. nanoparticles or more efficient bacterially expressed systems. The results further support the adoption of RNAi as an alternate technology to the conventional RPW controlling strategies.

## Supporting information

**S1 Raw data.**
(XLSX)

## Author Contributions

**Conceptualization:** Muhammad Naeem Sattar, Adel A. Rezk, Hamadttu Elshafie, Jameel M. Al-Khayri.

**Data curation:** Muhammad Naeem Sattar, Muhammad Nadir Naqqash, Adel A. Rezk.

**Formal analysis:** Allah Bakhsh.

**Funding acquisition:** Hamadttu Elshafie, Jameel M. Al-Khayri.

**Investigation:** Muhammad Naeem Sattar, Khalid Mehmood.

**Methodology:** Muhammad Nadir Naqqash, Khalid Mehmood.

**Resources:** Muhammad Nadir Naqqash, Jameel M. Al-Khayri.

**Software:** Muhammad Nadir Naqqash, Allah Bakhsh.

**Validation:** Muhammad Naeem Sattar, Muhammad Nadir Naqqash.

**Writing – review & editing:** Allah Bakhsh, Jameel M. Al-Khayri.

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
