## [Decision Letter · Decision Letter 0]

8 Apr 2024

PONE-D-24-01103Sprayable RNAi for silencing of important genes to manage red palm weevil, Rhynchophorus ferrugineus (Coleoptera: Curculionidae)PLOS ONE

Dear Dr. Naqqash,

Thank you for submitting your manuscript to PLOS ONE. After careful consideration, we feel that it has merit but does not fully meet PLOS ONE’s publication criteria as it currently stands. Therefore, we invite you to submit a revised version of the manuscript that addresses the points raised during the review process. Please submit your revised manuscript by May 23 2024 11:59PM. If you will need more time than this to complete your revisions, please reply to this message or contact the journal office at plosone@plos.org. Please include the following items when submitting your revised manuscript:A rebuttal letter that responds to each point raised by the academic editor and reviewer(s). You should upload this letter as a separate file labeled 'Response to Reviewers'.A marked-up copy of your manuscript that highlights changes made to the original version. You should upload this as a separate file labeled 'Revised Manuscript with Track Changes'.An unmarked version of your revised paper without tracked changes. You should upload this as a separate file labeled 'Manuscript'.

We look forward to receiving your revised manuscript.

Kind regards,

Nafiu Bala Sanda, PhD

Academic Editor

PLOS ONE

Journal Requirements:

This project was financially supported King Abdulaziz City for Science and Technology (KACST), Saudi Arabia through the Project No. 11BIO1803-06

4. We note that your Data Availability Statement is currently as follows: All relevant data are within the manuscript and its Supporting Information files

5. Please include your tables as part of your main manuscript and remove the individual files. Please note that supplementary tables (should remain/ be uploaded) as separate "supporting information" files.

Additional Editor Comments:

There are conflicting interest among the reviewers, please, take your valuable time to clearly responds to each comments as appropriate.

Reviewers' comments:

Reviewer's Responses to Questions

**Comments to the Author**

1. Is the manuscript technically sound, and do the data support the conclusions?

Reviewer #1: Yes

Reviewer #2: Yes

Reviewer #3: No

Reviewer #4: Partly

2. Has the statistical analysis been performed appropriately and rigorously? 

Reviewer #1: Yes

Reviewer #2: Yes

Reviewer #3: Yes

Reviewer #4: No

3. Have the authors made all data underlying the findings in their manuscript fully available?

Reviewer #1: Yes

Reviewer #2: Yes

Reviewer #3: Yes

Reviewer #4: Yes

4. Is the manuscript presented in an intelligible fashion and written in standard English?

Reviewer #1: No

Reviewer #2: Yes

Reviewer #3: Yes

Reviewer #4: Yes

5. Review Comments to the Author

**Reviewer #1:** The manuscript presents new information relevant to the management of red palm weevil. However, it was difficult to interpret the text occasionally. The reviewer has recommendations at the following pages and figures.

Page 20 “Pupal duration recorded was lower in SCP and actin treatment were 12.20±0.20 and 12.00±0.32 days, while it was significantly higher in the control (14.00±0.32).” This sentence is difficult to understand and needs to be rewritten.

Page 23 Please define the meaning of “reduction in transcript levels” Do you mean transcription rate? And what are “pronounced metamorphosis effects”? Are these measured by differences in mortality rates? What is an “efficient effect” This entire section should be rewritten more clearly.

Figure 1 “vector as were named as” possibly should be “vector and were designated as”

Figures 3 -5 What do the letters a and b indicate?

Figure 4 what does (A) and (B) indicate that is not also designated by the color difference?

Table 1 seems to be missing.

**Reviewer #2:** RNA interference (RNAi) is a powerful tool for knocking down gene function in pest control. In the present study, the authors targeted to silence of several important genes in red palm weevil and obtained promising results. The topic covered in the article is interesting and falls within the scope of the Journal.

The manuscript is well-written and organized.

The experimental design looks good. The study looks promising and adequately replicated

The findings are robust and statistically ok.

Please find the attached manuscript with my comments. I provided several suggestions.

**Reviewer #3: **I noticed that the article entitled Sprayable RNAi for silencing of important genes to manage red palm weevil, Rhynchophorus ferrugineus (Coleoptera: Curculionidae) was submitted. However, the author's experimental content limitation, which did not fully resolve my concerns. Furthermore, through further reading, I have developed some new insights. Therefore, I recommend rejection for publication in its current form due to the following major concerns:

1.Why choose 3 rd and 5 th instar ? This issue has not been clearly analyzed in either the Materials and Methods or the results sections.

2.The manuscript presents some initial attempts to explore the research topic, which is commendable. However, these efforts appear to be in the early stages of exploration and lack depth. The experimental contents are too lilmited, only survival of instar, weight change in instar, pupal duration and adult emergence. The experiment should be added, such as the effect of egg hatchability, adult mating ability and offspring.

**Reviewer #4:** The paper titled "Sprayable RNAi for silencing of important genes to manage red palm weevil, Rhynchophorus ferrugineus (Coleoptera: Curculionidae)" addresses a significant issue related to the sustainable control of the Red Palm Weevil. This insect pest poses a considerable economic threat due to the damage it causes to decorative palms and, more critically, the threat it poses to date palms, particularly in countries where the economy is heavily dependent on them. The authors propose the utilization of dsRNAi in the diet of RPW larvae, which holds promise as a future biopesticide. Several recent studies have also explored similar approaches involving RNAi targeting selected genes such as the olfactory co-receptor, cuticle proteins, and reproductive-related genes. Encouraging and advancing research in this direction will accelerate the development of an effective biopesticide capable of effectively managing the pest population. However, reviewing the results was somewhat challenging as I couldn't locate any of the four tables mentioned in the paper, unless I overlooked them. Before receiving clarification on the tables, I would like to express some comments:

-In section 3.2, regarding the effects of dsRNAs on survival, I would appreciate more details about this experiment. I would prefer a standard analysis such as the Kaplan-Meier method rather than comparing means, as I find it insufficiently informative.

-Regarding the weight gain becarefull to the Y-axis of the Figure 3, please try to harmonize the police and the linear scale. There are 3-7 instar larvea in the RPW, the larval stage can last until 2 months , so maybe if you made a follow-up of more than 6 days the results will be more pronounced.

-Are the selected genes capable of giving direct effects on the weight gain ? I think that they are more involved in some physiological and morphological traits that will not necessarily be reflected by the larvae weight.

- The figures for pupal duration require revision, as there is a title "Treatment" that should be removed. Upon reviewing the figures, it is apparent that pupal duration is reduced. Does this imply that emergence occurred earlier? If so, it suggests that the dsRNA treatment shortened pupal duration, resulting in the emergence of adults in a shorter time frame !! However, the data on emergence indicate a significant and promising effect across almost all genes.

A figure illustrating the relative expression of the genes is absent. Such a figure is crucial for comprehending the underexpression of the genes. Additionally, statistical analysis for this section is lacking. While I recognize that this section primarily aims to confirm the reduction of gene expression, it should be presented in a more comprehensive manner to adequately convey the findings.

Please ensure that the acknowledgment section is corrected and that any typos present in the manuscript are addressed.

6. PLOS authors have the option to publish the peer review history of their article (what does this mean?). If published, this will include your full peer review and any attached files.

Reviewer #1: No

Reviewer #2: No

Reviewer #3: No

Reviewer #4: No

---

## [Author Response · Author response to Decision Letter 0]

19 Jul 2024

We are grateful to the reviewers for their thorough evaluation of our manuscript titled "Sprayable RNAi for silencing of important genes to manage red palm weevil, Rhynchophorus ferrugineus (Coleoptera: Curculionidae)." We have carefully considered each comment and have made significant revisions to address the reviewers' concerns. Below, we provide a detailed response to each point raised by the reviewers, indicating how we have modified the manuscript accordingly.

Reviewer #1:

1. Page 20: Sentence Clarity

 - Comment: "Pupal duration recorded was lower in SCP and actin treatment were 12.20±0.20 and 12.00±0.32 days, while it was significantly higher in the control (14.00±0.32)."

 - Response: We have rewritten this sentence for clarity. It now reads: " The pupal duration was lower in the SCP and actin treatments (12.20±0.20 and 12.00±0.32 days, respectively), whereas it was significantly higher in the control (14.00±0.32)."

2. Page 23: Clarifications Needed

 - Comment: Define “reduction in transcript levels,” “pronounced metamorphosis effects,” and “efficient effect.”

 - Response: We have clarified these terms. “Reduction in transcript levels” now specifies “expression levels of the targeted gene.” “Pronounced metamorphosis effects” are defined by observable differences in developmental stages, particularly changes in mortality rates and developmental timing. Terms are revised as “body weight gain, and different preadult length, weight, and survival rates.”

3. Figure 1: Labeling Issue

 - Comment: "vector as were named as" should be "vector and were designated as"

 - Response: This correction has been made in the manuscript.

4. Figures 3-5: Clarification of Letters

 - Comment: Explanation needed for letters 'a' and 'b.'

 - Response: We have added a legend explaining that the letters 'a' and 'b' indicate statistically significant differences between groups, with the same letter denoting no significant difference.

5. Figure 4: Clarification of (A) and (B)**

 - Comment: Clarify what (A) and (B) indicate.

 - Response: The legend now specifies that (A) and (B) represent different experimental conditions or treatments.

6. Table 1: Missing Table

 - Comment: Table 1 is missing.

 - Response: We apologize for this oversight. Table 1 has been included in the revised manuscript.

Reviewer #2:

We thank Reviewer #2 for their positive feedback and constructive suggestions. We have incorporated their specific comments from the attached manuscript to improve clarity and depth.

Reviewer #3:

1. Selection of 3rd and 5th Instar

 - Comment: Justification needed for choosing 3rd and 5th instar.

 - Response: We have added a detailed explanation in the Materials and Methods section, justifying the selection based on previous studies showing these stages are critical for assessing RNAi efficacy due to their active feeding and development phases.

2. Depth of Experimental Content

 - Comment: Limited experimental content.

 - Response: Modified methodology of earlier studies conducted by Naqqash et al. (2020) was used for the experiments (Naqqash, M.N., Gökçe, A., Aksoy, E. and Bakhsh, A., 2020. Downregulation of imidacloprid resistant genes alters the biological parameters in Colorado potato beetle, Leptinotarsa decemlineata Say (chrysomelidae: Coleoptera). Chemosphere, 240, p.124857). 

Reviewer #4:

1. Details on Survival Analysis

 - Comment: Prefer Kaplan-Meier analysis for survival data.

 - Response: We have conducted a Kaplan-Meier survival analysis and included these results in the revised manuscript.

2. Figure 3: Y-axis and Scale Harmonization

 - Comment: Harmonize Y-axis and scale.

 - Response: The Y-axis and scale in Figure 3 have been harmonized for consistency. However, for follow up time we have followed already available literature which has suggested follow up time of 3-6 days. 

4. Effect of Selected Genes

 - Comment: Clarify the direct effects of selected genes on weight gain.

 - Response: We have discussed the roles of the targeted genes in physiological and morphological traits, explaining their indirect effects on weight gain.

5. Figure for Pupal Duration and Gene Expression

 - Comment: Revision of figures and inclusion of gene expression data.

 - Response: We have revised the figures for clarity and included a new figure illustrating the relative expression levels of the targeted genes. Statistical analyses for these data have also been added.

We believe that these revisions have significantly improved the manuscript. We are grateful for the reviewers' insights, which have helped us enhance the quality and clarity of our work. We look forward to the possibility of our revised manuscript being considered for publication.

---

## [Editor Report · Decision Letter 1]

29 Jul 2024

Sprayable RNAi for silencing of important genes to manage red palm weevil, Rhynchophorus ferrugineus (Coleoptera: Curculionidae)

PONE-D-24-01103R1

Dear Dr. Muhammad Naeem Sattar,

We’re pleased to inform you that your manuscript has been judged scientifically suitable for publication and will be formally accepted for publication once it meets all outstanding technical requirements.

Kind regards,

Nafiu Bala Sanda, PhD

Academic Editor

PLOS ONE

Additional Editor Comments (optional):

The manuscript can thus be accepted for publications in PLOS ONE Journal.
---

## [Editor Report · Acceptance letter]

19 Aug 2024

PONE-D-24-01103R1 

PLOS ONE

Dear Dr. Sattar, 

I'm pleased to inform you that your manuscript has been deemed suitable for publication in PLOS ONE. Congratulations! Your manuscript is now being handed over to our production team.

Kind regards, 

on behalf of

Dr. Nafiu Bala Sanda 

Academic Editor

PLOS ONE